# Exploring the Motives for Online Fashion Renting: Insights from Social Retailing to Sustainability

**Stacy H. Lee [1],\* and Ran Huang [2]**

[1]   Department of Hospitality and Retail Management, Texas Tech University, Lubbock, TX 79409, USA

[2]   Eskenazi School of Art, Architecture + Design, Indiana University, Bloomington, IN 47405, USA; huangran@iu.edu

\*   Correspondence: stacy.h.lee@ttu.edu; Tel.: +1-806-834-0091

**Abstract:** Despite the exponential growth of collaborative consumption practices, online fashion renting, an important type of collaborative fashion consumption, is still underexplored. Drawing on the theories of Reasoned Action (TRA) and Innovation Diffusion, we developed a holistic research framework to explore the motives for online fashion renting. By analyzing a total of 300 usable responses collected by a research market company using structure equation modeling (SEM), we found that attitudes and subjective norms positively influenced consumers' intentions to engage with online fashion rental services. Moreover, we found that environmental awareness also had a significant influence on attitudes toward fashion renting through online platforms, and that relative advantage, amplified by personal innovativeness and fashion consciousness, also positively influenced consumers' attitudes toward online fashion renting. Interestingly, price consciousness did not contribute to relative advantage.

**Keywords:**   online fashion renting; collaborative consumption; environmental awareness; relative advantage; motives

## 1. Introduction

Collaborative consumption is defined as economy sharing activities among consumers, or peer-to-peer commerce, that provides alternative venues for industries while lengthening the lifecycle of products [1,2]. Collaborative consumption offers a more environmentally friendly mode of consumption (e.g., renting, bartering, trading, lending, and swapping of goods), all of which involve the conservation of resources [3,4]. With temporary usage, customers cannot claim full property rights to products, which in turn leads to low or nonexistent levels of risks and responsibilities related to the products [5]. The terms "collaborative consumption," "sharing economy," and "access-based consumption," are used interchangeably when referring to the emerging phenomenon of the peer-to-peer (P2P) mode of production and consumption of products and services [6,7]. However, collaborative consumption is the more appropriate term to use for business-to-consumer services (B2C), such as Line, Zipcar, and Rent the Runway [2].

When it comes to the apparel marketplace, four types of collaborative consumption have been identified: temporary renting, subscription leasing, trading, and entrusting [8]. Despite utilizing different business models, they all support consumers' pursuit of resource efficiency and the reduction in waste [9]. Likewise, there has been exponential growth in the market for online fashion rental services, including well-known businesses such as the UK-originated Girl Meets Dress, U.S.-based Rent the Runway and LendMyTrend, and China-originated Meilizu [10]. Rent the Runway has made a significant impact, valued at over USD 1 billion in 2019 [11]. Girl Meets Dress and other U.K.-based clothing rental companies together contribute to a potential market value of GBP 923 million [12].

Online portals provide fashion rental services for a wide range of items, from outfits for special occasions (such as weddings and formal parties) to daily apparel and accessories. Customers can browse hundreds of styles through these portals to choose their desired outfits. Pre-paid shipping services are arranged for the delivery and return of rental items. Returned apparel is cleaned and maintained after every rental. However, the nature of collaborative apparel consumption might be different from that of collaborative consumption in other industry sectors such as automobiles, toys, and/or vacation home rentals—the former may meet consumers' hedonic interests, whereas the latter may satisfy their utilitarian needs [8]. Additionally, the majority of online fashion renting platforms (e.g., Girl Meets Dress, and Meilizu) employ the B2C model, whereas the service-based rental market (e.g., Uber, Lyft, and Airbnb) mainly uses a P2P approach [13]. Therefore, due to the fact that fashion renting is predominately established in a B2C model, this study aims to explore consumers' motivations for online renting within the apparel domain.

Rooted in social psychology, the Theory of Reasoned Action (TRA) is used extensively to capture consumers' decision-making processes in a variety of areas [14]. Previous literature has found that both consumers' attitudes toward online shopping and their online fashion renting behavior has a positive influence on their behavioral intentions [15,16]. Across different disciplines and industries, studies on collaborative consumption have investigated two critical factors, relative advantage and environmental considerations, which often lead to participation in collaborative consumption [17–21]. Moreover, the Innovation Diffusion Theory (IDT) identified the extrinsic motives for online fashion renting as relative advantage and found that there was a different significance given to motives such as personal innovativeness, fashion consciousness, and price consciousness [16,22,23]. On the other hand, intrinsic motives such as environmental consciousness and awareness play significant roles in collaborative consumption, as collaborative consumption increases the duration of apparel utilization and thus reduces clothing waste [2,24]. Similar but different from Tu and Hu's study [16], this study tries to investigate how consumers perceive fashion online renting services, whether as an environmental aspect or for fulfilling fashion-oriented self-interest. Despite the fact that collaborative fashion consumption has become a noteworthy phenomenon, knowledge about how consumers perceive engaging in this practice online and the important motivating factors remains relatively limited. In addition to recognizing how online fashion rental services can provide better advantages by appealing to extrinsic motives, it is also important to understand environmental awareness as an intrinsic motive for pursuing online fashion rental services [2,24]. In order to take a holistic view of the psychological motivations for online fashion renting, it is imperative to investigate both extrinsic and intrinsic perspectives. As previous literature has emphasized either an intrinsic or an extrinsic view [2,18–21,24], this study aims to fill a gap by developing a comprehensive framework that incorporates both aspects. Therefore, there are two objectives of this study: (1) to investigate the TRA as applied to online fashion renting; (2) to explore both the intrinsic and extrinsic motives for pursuing online fashion renting.

Theoretically, the current research will enhance the collaborative consumption literature by extending the existing knowledge base on the TRA and IDT with a study that depicts both the intrinsic and extrinsic motives for participating in online fashion rental services. Moreover, the results of this study can allow newly emerging fashion rental businesses to understand the genuine drivers of consumers' intentions. Marketing for online fashion rental services could either target an individual's innovativeness or emphasize the environmental benefits inherent in this practice. Service providers should also pay attention to social influences by incorporating the effect of peer validation into their marketing communications.

## 2. Literature Review

### 2.1. Online Renting in the Fashion Context

Collaborative consumption has been extensively studied in various settings such as car sharing [25], bike sharing [26], and accommodation sharing [27]. The extant literature investigates collaborative fashion consumption, which can be explored from three main perspectives: the environment, the business, and the consumer. Essentially, the environmental impact of collaborative fashion consumption is a key topic that has drawn considerable attention from the academy [28]. By introducing a typology of collaborative fashion consumption from the environmental perspective, Iran and Schrader [29] attributed the positive environmental impact of collaborative fashion consumption to the increased utilization of garments and the reduced consumption of new clothing. Therefore, environmentally conscious consumers are interested in sustainable apparel services such as online garment rentals. Park and Armstrong [8] examined consumer behavior in the collaborative apparel consumption framework. Whereas the authors maintained that collaborative consumption of apparel products has been hindered due to its symbolic nature, they also suggested that political consumerism, defined as "the consumer making a consumption choice based on their personal ideology" [8] (p. 471), and convenience encourages participation. Johnson, Mun, and Chae [30] empirically illustrated that consumers' integrity and previous offline experiences are antecedents to their attitudes toward online collaborative apparel consumption. In turn, attitudes, subjective norms, and offline experiences exert positive influences on intentions toward engaging in collaborative consumption of apparel products. With advances in information and communication technologies, the procedures involved in collaborative consumption can be carried out online and thus streamlined, which facilitates the increased utilization of collaborative consumption. Therefore, consumers' intentions to shift the focus from individual private ownership of products to collaborative consumption can lead to enhanced value from a social and environmental perspective. In this regard, previous studies [31] have also indicated that factors around self-interest, such as lower cost, are additional motivators for participating in collaborative consumption.

### 2.2. TRA in Online Fashion Renting

As a theoretical foundation that examines the rational and cognitive components involved in the process of consumer decision making, the TRA has been applied in diverse contexts in which individuals have volitional control, and it can predict behaviors and willingness towards behaviors that are determined by attitudes and social norms [15]. Therefore, a number of empirical studies have found that positive attitudinal responses toward online shopping result in stronger intentions to shop [32], purchase green products [33], and donate secondhand clothes [34]. The TRA assumes that intentions lead to specific behaviors by apprehending fundamental motivational factors. In this sense, the immediate determinant of the actual behavior is intention, which refers to "people's expectancies about their own behavior in a given setting" [35] (p. 288) and can be determined by attitudes and subjective norms. More specifically, consumers tend to become involved in collaborative consumption practices that allow them to use products for a certain period of time without claiming ownership [36]. Likewise, purchase intentions of fashion renting through online platforms indicates the likelihood that the consumer would prefer to rent products online. In other words, since behavioral intentions have a close relationship with actual behavior, the intensity of the behavioral intention is likely to be interpreted as a manifestation of the actual behavior, such as purchasing ethical products and renting fashionable clothes [16,37].

To better capture the behavioral intentions behind online fashion renting, two key factors were explored: (1) attitudes towards the action; (2) subjective norms [38]. Attitude reflects the degree to which an individual has either a favorable or unfavorable evaluative assessment of an action [37]. A positive attitude towards online fashion renting indicates that the consumer would perceive utilizing online rental services for fashion items as beneficial and enjoyable. The positive impact

of attitude on behavioral intention has been well documented in a variety of business domains including green product consumption [33], food consumption [39], mobile data service adoption [40], e-commerce activities [41], pro-environmental consumer behavior [42], and so on. Within the context of collaborative consumption, empirical evidence also supports this relationship, suggesting that attitudes toward online fashion renting have a positive influence on behavioral intentions [16]. On the other hand, the term "subjective norms" is defined as an individual's perception of the pressure exerted by the social environment that pushes or restrains him/her from taking certain actions [43] (p. 37). Prior research has demonstrated that subjective norms play a significant role in recycling behavior [44], local food consumption [45], adopting self-service technologies [46], online information seeking [47], and online purchasing behavior for clothing [48]. Social influences (e.g., word of mouth, and referrals from close friends and family) may also be highly significant in shaping one's intentions toward certain behaviors, especially for fashion-related products and services [15,30].

Although online fashion renting is viewed differently from online shopping in terms of its business model, the former engages a similar process that involves searching for products and engaging in online transactions [17]. Research has found that attitudes and subjective norms have a positive impact on intentions to use online rental services, especially for consumers who are experienced in collaborative apparel consumption [30]. Therefore, we employed the TRA to better capture online fashion renting, and the following hypotheses were formed. In the setting of online fashion renting,

**Hypothesis 1 (H1).** *Attitudes positively impact intentions to use online fashion rental services.*

**Hypothesis 2 (H2).** *Subjective norms positively impact intentions to use online fashion rental services.*

### 2.3. Intrinsic versus Extrinsic Motivations towards Online Fashion Renting

The extant literature on collaborative consumption suggests that economic and environmental considerations are the two main factors prompting consumers to become involved in collaborative consumption [17]. Environmental awareness is a significant driver of consumers' willingness to purchase sustainable products [1,15,49]. Similarly, environmental awareness plays an important role in consumers' choices of environmentally friendly products [20]. Consumers are willing to put extra effort into engaging in environmentally friendly practices even if doing so sacrifices convenience, if they determine that it is important to protect the environment. This finding indicates that consumers' intrinsic motives for practicing environmental consciousness would likely have a positive effect on their decisions to pursue online fashion renting. As an innovative business model, online fashion renting enables the sharing of products and services through online platforms. Originating in the field of sociology, the Innovation Diffusion Theory (IDT) recognizes that there are five attributes which affect attitudes towards an innovation: relative advantage, compatibility, complexity, trialability, and observability, with the first two being considered the most impactful factors [50,51]. Relative advantage and outcome expectations are identified as the most prominent factors in predicting behaviors [52], product or service innovations [21,53], and using services such as Airbnb [19]. As relative advantage reflects the benefits of utilizing collaborative consumption online, Tu and Hu [16] addressed the importance of highlighting the extrinsic motives contributing to online renting's relative advantages. Compared to other components of the IDT, relative advantage offers a broader perspective that highlights important advantages of the overall product/service performance [19]. Namely, perceived relative advantage better reflects individual consumers' overall evaluations of a certain product/service, as compared to other components of IDT. Thus, this study focuses on relative advantage in order to capture a comprehensive picture of consumers' perceptions of online fashion renting.

### 2.4. Environmental Awareness

Collaborative consumption has a positive impact on the environment because it involves sharing products, accommodations, or transportations without ownership [19,36,54]. Prior research has

acknowledged that consumers' perceived sustainability of collaborative consumption positively affected their attitudes towards this practice [55]. Tussyadiah and Pesonen [24] supported the view that sustainability benefits are essential motivators for pursuing accommodation sharing. Likewise, the environmental benefits of collaborative fashion consumption were found to be significant, as it increases the apparel utilization rate and reduces waste from clothing disposal [2]. Therefore, it is expected that perceptions of the environmental (sustainability) benefits of collaborative consumption will shape attitudes towards this practice. Gam [56] demonstrated that consumers with pro-environmental mindsets tend to spend money on protecting the environment. Moreover, Gam [56] also found a positive association between such attitudes and sustainable fashion consumption. Even if practicing environmentally conscious behavior is inconvenient, environmentally conscious individuals still prefer to purchase sustainable products or/and to pursue environmentally friendly behaviors [49]. Accordingly, if consumers are highly environmentally aware, they may react positively towards online fashion rental services. To understand environmental awareness as an important intrinsic motive, the following hypothesis was posited: In the setting of online fashion renting,

**Hypothesis 3 (H3).** *Environmental awareness positively impacts attitudes towards online fashion rental services.*

### 2.5. Relative Advantage in Online Fashion Renting

Relative advantage is defined as the degree to which an innovation is able to outperform other state-of-the-art ideas [51]. If individuals recognize the relative advantages or benefits gained from performing a behavior, they are likely to react favorably toward the behavior [57]; in contrast, if individuals assess that there are more disadvantages to performing a behavior, they may have an unfavorable attitude toward this behavior. Empirical studies support the importance of relative advantage or perceived usefulness as a predictor of attitudes toward online shopping [58]. The TRA and IDT are proposed in different disciplines such as mobile applications [59] yet share the view that adoption of online shopping is determined by its perceived attributes [60]. Moreover, online fashion renting provides similar advantages over typical online purchasing in many aspects, such as: (1) users gain access to desirable fashion items and even designer products at affordable prices, and (2) users are able to alter their wardrobe more often [28,61]. With limited rights to the rentals, consumers bear zero or low risks and responsibilities related to possession of the products [62]. Since online fashion renting is an innovation that falls under the framework of the IDT, the following hypothesis was proposed: In the setting of online fashion renting,

**Hypothesis 4 (H4).** *Relative advantage positively impacts attitudes towards online fashion rental services.*

### 2.6. The Relationship of Motivation to Relative Advantage

Previous literature has argued that the higher the relative advantage involved in online collaborative consumption, the higher the likelihood of adopting innovations such as online fashion renting [16]. Similarly, numerous studies have stressed that cost-savings and utility maximization are dominant motivators for participating in collaborative consumption [27]. Therefore, it is important to understand the determinants of relative advantage [16]. These authors found personal innovativeness as one of the key factors that influences attitudes toward certain behaviors within the contexts of B2C car sharing services and the consumer-to-consumer (C2C) online community accommodation marketplace. Moreover, previous literature found that more fashion-conscious individuals tend to have positive attitudes towards sustainable fashion consumption and status consumption [23].

### 2.7. Personal Innovativeness

Personal innovativeness is associated with an individual's willingness to change [16]. Personal innovativeness is defined as a willingness to take risks and engage in innovative behaviors [2,16,60].

Jones, Sundaram, and Chin [63] emphasized the importance of personal innovativeness in forming consumers' attitudes toward new systems. As an important personality trait influencing an individual's adoption of innovations, personal innovativeness has a positive impact on the perceived relative advantage of wearable technologies [64]. Based on the Theory of Planned Behavior, Limayem, Hirt, and Cheung [65] found that personal innovativeness has a positive influence on both attitudes and behavioral intentions towards online shopping. Similarly, Tu and Hu [16] found that personal innovativeness positively influences attitudes toward online fashion renting. Based on these findings, consumers with greater levels of personal innovativeness are more likely to perceive online fashion renting with more relative advantage, because these individuals have positive attitudes toward innovations [60]. Consequently, the following hypothesis was posited: In the setting of online fashion renting,

**Hypothesis 5 (H5).** *Personal innovativeness positively impacts perceptions of its relative advantages.*

*2.8. Fashion Consciousness*

Fashion consciousness refers to individuals' "desire for and adoption of up-to-date styles to maintain one's status in a social network" [22] (p. 1410). In the dynamic and fast-paced fashion market environment, collaborative consumption is deemed as an increasing phenomenon, which greatly influences consumers' decision making [54]. According to Dutta-Bergman and William [66], individualistic consumers are more fashion conscious and care about their own lives more than those of others. As consumption behavior is associated with social identity, consumers who are likely to pursue trendy products are also likely to choose collaborative consumption over ownership [5]. This is because a greater number of fashion-conscious consumers were shown to adopt innovative and fashionable products [22,27]. Thus, fashion-conscious consumers may perceive that online fashion renting can fulfil their desires to be up-to-date and fashionable [2]. Consequently, the following hypothesis was proposed: In the setting of online fashion renting,

**Hypothesis 6 (H6).** *Fashion consciousness positively impacts perceptions of its relative advantages.*

*2.9. Price Consciousness*

Price consciousness refers to "a cognitive tradeoff between the perceived benefits of the offering and specific monetary cost for using it" [67] (p. 225). Empirical evidence has supported that economic benefit is an important determinant in the choice to engage in collaborative consumption. Mohlmann [27] identified saving money and maximizing utility are as two crucial incentives for collaborative consumption. However, other findings about the influence of price consciousness on collaborative consumption have been reported in the tourism and car rental industries [67]. Tussyadiah and Pesonen [24] stressed the significance of the cost-saving features of collaborative consumption in the use of P2P accommodation sharing. However, a study by Mohlmann [27] also found that cost saving was not a significant factor in increasing satisfaction from car-sharing or accommodation-sharing. Although there were polarizing results in terms of economic benefits, price consciousness is still considered to be one of primary factors involved in deciding to adopt online fashion renting [15], which in turn may transfer as a relative advantage. For instance, scholars found that online fashion rental services enable consumers to update their wardrobes more often at a reasonable cost [61]. Consequently, the following hypothesis was proposed: In the setting of online fashion renting,

**Hypothesis 7 (H7).** *Price consciousness positively impacts perceptions of its relative advantages.*

Based on the literature review, the following conceptual research model is proposed (Figure 1):

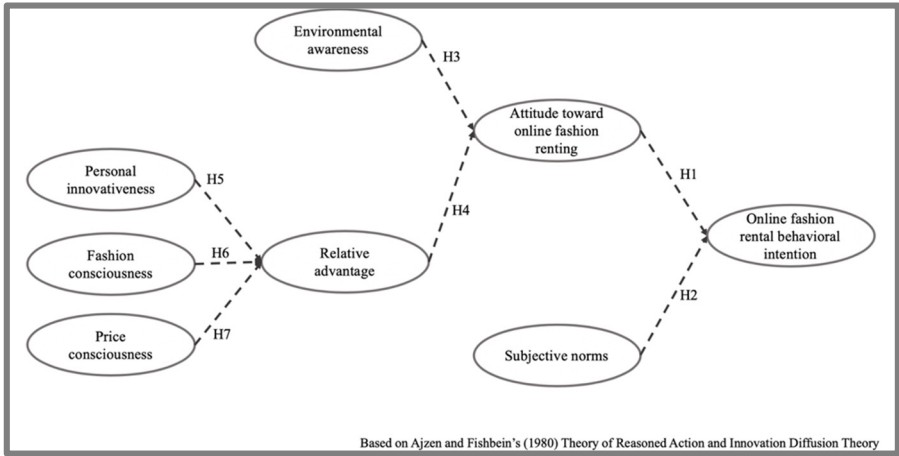

**Figure 1.** Research framework of this study.

## 3. Results

### 3.1. Participants and Procedures

Data were collected through an online survey in 2018, and a total of 300 usable responses were obtained from an established research company (see Table 1 for the respondents' characteristics). The target sample frame consisted of U.S. consumers over 18 years old. Quota sampling was adopted to capture perceptions from different genders and age groups, with a similar composition across groups. To help them understand online fashion rental services, all participants were asked to read a short summary describing online fashion rental prior to taking the main survey. This summary explained the types of rental services offered for a wide range of fashion items, from outfits for special occasions (such as weddings and formal parties) to daily fashionable apparel and accessories, and the process of using the online fashion rental portal from renting to returning. In order to examine the difference between users and non-users of online fashion rental services, the Levene's Test of Homogeneity of Variance between non-users and users for all the constructs was conducted, which indicated a *p*-value greater than 0.50, suggesting that the variances in these two samples could be considered as equal. Thus, further analyses were conducted without dividing the users from the non-users of online fashion rental services.

### 3.2. Measurement Scale

To ensure the content validity of the instruments, measurement items from the literature were employed with minor modifications to adapt to the context of this study. The wording of the measurement scale was modified to fit in the research context, namely, fashion retail businesses. Moreover, online fashion rental intentions were measured with two items from Karahanna, Straub, and Chervany [68]. Attitudes were evaluated with five items from Ajzen [57], while subjective norms were assessed with four items adapted from Ajzen [57] and Ozaki [69]. Perceived relative advantage was examined with a four-item scale from Karahanna et al. [68]. Three items were used to measure price consciousness [70], whereas two items were used to measure fashion consciousness [22]. Three items assessing personal innovativeness [60] and three items measuring environmental awareness [71] were also adopted. According to Gam [56] and Han and Yoon [71], environmental awareness reflects general perceptions of the importance of selecting products and services based on their environmental benefits, which shapes attitudes towards eco-friendly practices including collaborative consumption. Thus, environmental awareness was measured using items that indicated consumers' general concerns for the environment. Apart from the above constructs, respondents' demographic information was also collected. Respondents were also asked whether or not they had previous fashion rental experience.

To mitigate the possible effects of common method variance (CMV), the following procedures were conducted in designing the set of questionnaires in accordance with the recommendations in the literature [72]. First, to reduce the scale endpoint commonalities and anchoring effect, different endpoints and formats of the measurement scale were utilized for the research variables. To be specific, a seven-point sematic differential scale was adopted for measuring attitudes (e.g., 1: harmful, 7: beneficial), while online fashion rental intentions were assessed on a seven-point Likert scale (1: extremely unlikely, 7: extremely likely). Other constructs were tested on a five-point Likert scale (1: strongly disagree, 5: strongly agree). To control for CMV, a marker variable was included in the set of questionnaires for ex post statistical analysis [73]. This marker variable is theoretically unrelated to the research variables. Ideally, it should also be disposed to the same causes for CMV as the main constructs by stimulating similar cognitive processes or response tendencies [74]. From this perspective, the marker variable that assessed respondents' attitudes towards social network usage [75] was used with a five-item sematic-differential scale, similar to the format of the measurement items for the main constructs such as attitude.

A rigorous process was followed to ensure content validity and construct reliability. The questionnaire was first prepared and reviewed to ensure the face validity of the constructs being assessed. Two university professors in related disciplines and a native English speaker who is considered a relevant sample in this study were invited to review the items. No difficulties associated with the instructions or the wording of the questionnaires were reported. All reviewers understood the instructions clearly and we found no issues relating to measurement items.

**Table 1.** Demographic characteristics.

|  | Frequency | % |  | Frequency | % |
|---|---|---|---|---|---|
| **Sample (N = 300)** |  |  |  |  |  |
| Gender |  |  |  |  |  |
| Female | 152 | 51% | Less than high school | 9 | 3% |
| Male | 148 | 49% | High school graduate | 117 | 39% |
|  |  |  | College or bachelor's degree | 138 | 46% |
| Age |  |  | Master's degree | 29 | 10% |
| 18–25 | 72 | 24% | Doctorate or professional degree | 7 | 2% |
| 26–35 | 75 | 25% |  |  |  |
| 36–45 | 75 | 25% |  |  |  |
| 46–55 | 78 | 26% | Office worker—Junior level [1] | 30 | 10% |
|  |  |  | Office worker—Managerial | 27 | 9% |
| Annual income |  |  | Manual worker [2] | 19 | 6% |
| Less than USD 10,000 | 61 | 20% | Front-end service provider [3] | 23 | 8% |
| USD 10,000–29,999 | 82 | 27% | Professionals [4] | 45 | 15% |
| USD 30,000–59,999 | 92 | 31% | Self-employed | 32 | 11% |
| USD 60,000–99,999 | 45 | 15% | Student | 31 | 10% |
| USD 100,000–149,999 | 17 | 6% | Retired | 6 | 2% |
| USD 150,000 or over | 3 | 1% | Unemployed | 50 | 17% |
|  |  |  | Others | 37 | 12% |
| Previous fashion rental experience |  |  |  |  |  |
| Yes | 71 | 24% |  |  |  |
| No | 229 | 76% |  |  |  |

Notes: [1] e.g., administrative/clerical; [2] e.g., worker in factory, construction, mechanic; [3] e.g., salesperson, waiter; [4] e.g., lawyer, doctor, teacher, etc.

### 3.3. Analysis and Findings

Assessment of Common Method Variance

To detect possible CMV, confirmatory factor analysis (CFA) [73] was conducted. A number of models were constructed, and their respective model fits were compared. To be specific, CFA was performed with the marker variable added and covaried with all constructs in the proposed measurement model. This model achieved a good fit ($\chi^2$ (369) = 647.943, $p < 0.001$, $\chi^2$/df = 1.756,

RMSEA = 0.050, IFI = 0.953, TLI = 0.945, and CFI = 0.953). The convergent and discriminant validity of all variables were also established. A second model was constructed by adding a common latent factor (CLF) that connected all observed items including marker variables. A chi-square difference test was conducted between the unconstrained model and the zero-constrained CLF models, which suggested that the two models were not significantly invariant (chi-square difference = 135.371, df = 30, $p < 0.001$). Therefore, the response bias was significantly different from zero, which indicated the existence of CMV. To further assess response bias, an additional model (referred to as the equal-constrained CLF model) was tested. This model was similar to the zero-constrained CLF model, yet with all factor loadings between the marker variable latent factor and manifest items constrained to be equal. The chi-square difference test between the unconstrained CLF model and the equal-constrained CLF model indicated that both were significantly different from each other (chi-square difference = 130.122, df = 29, $p < 0.001$). In other words, the response bias was found to be unevenly distributed across constructs. Consequently, the imputation of factor scores including the marker variable in the measurement model was conducted and CMV-adjusted variables were created for further structural equation model analysis.

*3.4. Hypothesis Testing*

To test the proposed hypotheses, this study conducted the relevant analyses in two steps, following Anderson and Gerbing [76]. First, a confirmatory factor analysis (CFA) was performed to assess the measurement model with maximum likelihood estimation (AMOS24). The CFA result indicated that the model yielded a good fit: $\chi^2$ (369) = 647.943, $p < 0.001$, $\chi^2$/df = 1.756, RMSEA = 0.051, IFI = 0.953, TLI = 0.945, and CFI = 0.953. All coefficients were significant. Fashion consciousness and online fashion renting intentions were elicited with two items as constructs. Hair, Babin, and Krey [77] argued that if the two-item constructs are combined into a model that consists of several other constructs with multiple items each, the overall model can be identified. The constructs had composite reliability scores ranging from 0.793 to 0.926 (see Table 2). Furthermore, with the average variance extracted (AVE) of each construct (>0.50), convergent validity was confirmed [78]. In addition, the AVE of each construct exceeded the squared correlation coefficients between associated pairs of constructs and thus discriminant validity of the constructs was supported (see Table 3).

**Table 2.** Results of exploratory and confirmatory factor analyses.

|  | Standardized Estimate |
| --- | --- |
| Environmental Awareness (AVE = 0.750, CR = 0.900) | |
| The effects of pollution on public health are worse than we realize. | 0.821 |
| Over the next several decades, thousands of species will become extinct. | 0.928 |
| Claims that current levels of pollution are changing earth's climate are exaggerated. | 0.845 |
| Personal Innovativeness (AVE = 0.562, CR = 0.793) | |
| If I heard about a new product/service, I would look for ways to experiment with it. | 0.743 |
| Among my peers, I am usually the first to try out new products/services. | 0.714 |
| I like to experiment with new products/services. | 0.789 |
| Fashion Consciousness (AVE = 0.799, CR = 0.888) | |
| I usually have one or more outfits of the newest style. | 0.931 |
| I keep my wardrobe up to date with the changing fashions. | 0.855 |
| Price Consciousness (AVE = 0.598, CR = 0.816) | |
| The money saved by finding low prices is usually not worth the time and effort. * | 0.727 |
| I would never shop at more than one store to find low prices. * | 0.717 |
| The time it takes to find low prices is usually not worth the effort. * | 0.867 |
| Perceived Relative Advantage (AVE = 0.626, CR = 0.869) | |
| Renting fashion items online would enable me to get apparel I want more quickly. | 0.802 |

**Table 2.** *Cont.*

| | Standardized Estimate |
|---|---|
| Renting fashion items online would enhance my effectiveness in getting the apparel I want. | 0.820 |
| Renting fashion items online would enable me to get apparel I want more easily. | 0.836 |
| Renting fashion items online would enable me to get apparel I want more cheaply. | 0.698 |
| Attitude (AVE = 0.693, CR = 0.918) | |
| Harmful–Beneficial | 0.713 |
| Pleasant–Unpleasant * | 0.858 |
| Good–Bad * | 0.910 |
| Worthless–Valuable | 0.799 |
| Enjoyable–Unenjoyable * | 0.869 |
| Subjective Norm (AVE = 0.717, CR = 0.883) | |
| Most people who are important to me think that I should rent fashion items online. | 0.706 |
| Most people who are important to me rent fashion items online. | 0.892 |
| The people in my life whose opinion I value rent fashion items online. | 0.926 |
| Online Fashion Renting Intention (AVE = 0.863, CR = 0.926) | |
| I intend to rent/continue to rent fashion items online within the next six months. | 0.939 |
| During the next six months, I plan to experiment with or regularly rent fashion items online. | 0.919 |
| Attitude toward Social Network (AVE = 0.631, CR = 0.893) | |
| Fun–Frustrating | 0.906 |
| Pleasant–Unpleasant * | 0.905 |
| Negative–Positive | 0.700 |
| Foolish–Wise | 0.599 |
| Enjoyable–Unenjoyable * | 0.816 |

Note: * indicates as reversed coded. AVE = Average Variance Extracted; CR = Composite Reliability.

**Table 3.** Convergent and discriminant validity check.

| | EA | PI | FC | PC | PR | AT | SN | INT | AT_SN |
|---|---|---|---|---|---|---|---|---|---|
| Environmental awareness (**EA**) | **0.750** | | | | | | | | |
| Personal innovativeness (**PI**) | 0.106 | **0.562** | | | | | | | |
| Fashion consciousness (**FC**) | 0.094 | 0.391 | **0.799** | | | | | | |
| Price consciousness (**PC**) | 0.008 | 0.008 | 0.014 | **0.598** | | | | | |
| Relative advantage (**PR**) | 0.163 | 0.234 | 0.195 | 0.003 | **0.626** | | | | |
| Attitude (**AT**) | 0.226 | 0.061 | 0.102 | 0.000 | 0.314 | **0.693** | | | |
| Subjective norm (**SN**) | 0.200 | 0.121 | 0.160 | 0.181 | 0.177 | 0.150 | **0.717** | | |
| Online fashion renting intention (**INT**) | 0.233 | 0.206 | 0.275 | 0.048 | 0.255 | 0.276 | 0.487 | **0.863** | |
| Attitude toward social network (**AT_SN**) | 0.092 | 0.074 | 0.060 | 0.000 | 0.071 | 0.187 | 0.023 | 0.061 | **0.631** |

*Note.* The numbers in the diagonal line are the average variance extracted by each construct. The numbers above the diagonal show the squared correlation coefficients between the construct.

Next, a structural equation model (SEM) was performed with maximum likelihood estimation, and the results showed a good model fit: $\chi^2$ (387) = 781.642, $p < 0.001$, $\chi^2$/df = 2.020, RMSEA = 0.058, IFI = 0.934, TLI = 0.925, and CFI = 0.933. The results supported all the hypotheses except for Hypothesis 7. Specifically, environmental awareness had a positive impact on attitude. Both personal innovativeness and fashion consciousness significantly contributed to perceived relative advantage, whereas price consciousness barely strengthened perceived relative advantage. Moreover, perceived relative advantage had a positive impact on attitude. Together, attitudes and subjective norms led to online fashion renting intentions. This model explained 36.7%, 27.7%, and 49.7% of the variances in attitude, perceived relative advantage, and online fashion renting intentions, respectively (Table 4).

**Table 4.** Results of the structural path model.

| | Path | Standardized Regression Coefficient (Beta) | *p*-Value |
|---|---|---|---|
| | *Hypothesis* | | |
| H1 | Attitude → Online fashion renting intention | 0.304 | *** |
| H2 | Subjective norm → Online fashion renting intention | 0.607 | *** |
| H3 | Environmental awareness → Attitude | 0.325 | *** |
| H4 | Perceived relative advantage → Attitude | 0.449 | *** |
| H5 | Personal innovativeness → Perceived relative advantage | 0.355 | *** |
| H6 | Fashion consciousness → Perceived relative advantage | 0.232 | ** |
| H7 | Price consciousness → Perceived relative advantage | −0.006 | n.s. |
| | *Control Variable Effect* | | |
| | Attitude toward social network → Online fashion rental intention | 0.033 | n.s. |

Note: *** $p < 0.001$, ** $p < 0.05$, * $p < 0.01$, n.s. = not significant.

## 4. Discussion

As there is a strong need for sustainable consumption in many different industries and areas of business, collaborative consumption has been one of the businesses models to facilitate the growth of sustainable consumption in the fashion industry over recent decades. The Theory of Reasoned Action and the Innovation Diffusion Theory were used to help understand the intrinsic and extrinsic motives for participating in online fashion renting. The current research extends Ajzen and Fishbein's TRA [14] to the setting of online fashion renting, which postulates that consumers' intentions to perform certain actions are shaped through psychological cognitive processes via both attitudes and subjective norms. To be specific, our findings suggest that online fashion renting attitudes optimistically influence behavioral intentions, which aligns with prior research in the context of online fashion rental services [15,30]. Online fashion renting intentions are also positively impacted by subjective norms. This result may be due to the fact that a predominant number of respondents (76%) reported having no experience with fashion rental services. When individuals face unfamiliar activities or innovations (e.g., online fashion renting), it may be natural to obtain information from close friends, family, or those closely related within their social circle [15].

Furthermore, environmental awareness and perceived relative advantage were found to be two significant drivers of consumer attitudes toward online fashion renting. As relative advantage means how much the benefits of an innovation are perceived to be better than existing ideas or practices, a higher level of perceived relative advantage will encourage consumers to participate in online fashion renting. Our results also support the view that relative advantage can be highly influential in increasing positive attitudinal responses toward online fashion renting. When consumers are aware of environmental issues such as pollution, positive attitudes toward online fashion renting are more likely to be evoked. Interestingly, in comparison with environmental awareness, relative advantage was found to be more effective in increasing positive attitudes towards online fashion renting. Perceived relative advantage was positively impacted by personal innovativeness and fashion consciousness, whereas price consciousness had no influence on perceived relative advantage. If consumers are open to innovation and sensitive to fashion trends, they may perceive that online fashion renting could provide more relative advantages. This result was somewhat expected due to the findings in previous literature [23]. It is because consumers mainly focus on the fun and enjoyment they experience from shopping, whereas the price rarely impacts on determining consumers' intentions to try fashion renting [79]. On the other hand, consumers who were sensitive to pricing did not consider that online fashion renting services could provide financial benefits to their consumption behavior. This might be because less than 25% of respondents had experience with online fashion renting. Additionally, along with fast-paced fashion trends, some consumers may perceive collaborative consumption to be too costly for their lifestyle, as many fashion rental services focus on special occasions and event-specific garments [23].

### 4.1. Theoretical Implications

The findings of this study enrich the collaborative consumption literature by accentuating that both theories contribute to the understanding of the intrinsic and extrinsic motives involved in adopting online fashion renting from consumers' perspectives. According to the Innovation Diffusion Theory, this study focused on one of the five attributes involved in learning about online fashion renting services. Previous literature has posited that relative advantage is the most important motivation for individuals to adopt innovations [15,27]. In fact, the current research demonstrates that environmental awareness is also considered a critical motive for innovation adoption, which expands the scope of knowledge on online fashion renting. More importantly, this study is one of the few to examine the different motives that trigger perceptions of the relative advantages of using online fashion rental services. To understand the determinants of perceived relative advantage in online fashion renting, three motivations were examined. Previously, these three motivations (personal innovativeness, fashion consciousness, and price consciousness), were found to be key factors that stimulate participation in fashion consumption as they are related to status consumption and sustainable consumption [23,27]. Within the context of online fashion renting, perhaps surprisingly, our findings suggested that personal innovativeness and fashion consciousness play important roles in consumers' perceived relative advantage of such business practices, which reflects the unique psychological characteristics of consumers in the online apparel rental market. Unlike other sustainable consumption practices [23], consumers who value innovation and fashion are more likely to perceive online fashion renting as having advantages; however, consumers who are price conscious can barely perceive those relative advantages from online fashion rental services.

### 4.2. Practical Implications

As our results demonstrate the importance of subjective norms in intentions to participate in online fashion renting services, managers and retail businesses should exploit relevant strategies to enhance social influence. For example, marketers could utilize key opinion leaders who may have positive reputations for sustainability in fashion-related contexts. In marketing to environmentally conscious individuals, it may be better to share information regarding the positive impact of online fashion rental practices from the environmental perspective. Consequently, testimonials from those in one's close social circle or network could be effective in learning about the nature of the collaborative consumption business model [80]. By examining both the intrinsic and extrinsic motives involved in forming attitudes, the findings of this study support that both types of motives are important in shaping positive attitudes towards online fashion renting. Consumers who have greater environmental awareness may have more positive attitudes towards online fashion renting services, as they would perceive that collaborative consumption can result in garment waste reduction, less resource usage of water and materials from production, and a smaller carbon footprint [55]. This result suggests that retail businesses should underscore how collaborative consumption relates to sustainability. For instance, online fashion renting businesses, such as Rent the Runway and Girl Meets Dress, have heavily emphasized sustainability, which has resulted in higher sales than companies using traditional fashion business models [10]. Moreover, managers could consider contributing a small percentage of sales to support environmentally conscious organizations. In this way, environmentally conscious individuals may perceive their participation in collaborative consumption as an action to help environmentally responsible or sustainable organizations.

Given the importance of personal innovativeness and fashion consciousness in online fashion renting, rental service providers could incorporate relevant features in their marketing communications to enhance their images as being innovative and fashionable. For example, Rent the Runway has recently utilized machine learning to identify individuals' fashion preferences and give consumers a unique renting experience, which thus increases consumers' perceptions of the advantages of using the company's service [81]. Additionally, fashion rental companies could create an online buzz with hashtags on social media to encourage consumers to share their experiences; such word-of-mouth

(WOM) activities to some extent shape subjective norms in the digital era and can enhance other consumers' intentions towards online fashion renting.

*4.3. Limitations and Future Research Directions*

There are some limitations which call for further investigation. First, this study only surveyed consumers in the U.S., which neglects online fashion renting platforms in different countries and cultures. Second, the sample used in this study was not equally distributed between those experienced with online fashion renting and those without experience, which might lead to a potential attitude–behavior gap. Lastly, the present study only investigated the relative advantage of online fashion renting, whereas some possible disadvantages might exist, such as hygiene-related issues with used clothing. These limitations invite future research opportunities. Future research can look into other possible determinants of intentions to use online rental services. For example, enhanced with advanced technologies such as machine learning, online renting behavior may be triggered by the hedonic and functional features of the rental platforms. Thus, future studies could examine consumers' perceptions of enjoyment and usefulness drawn from the technology acceptance model (TAM). As price consciousness was not supported as a factor in determining higher relative advantage for online fashion renting, future research could consider examining price factors in different online rental platforms that include fashion rental services. Additionally, the sample can be expanded to compare consumers who have experience with online fashion renting with those who have no experience. In this way, better insights can be gleaned as to appropriate marketing and promotion tactics to use for these two different target groups. For example, further investigation can look into the drivers which lead consumers to become users of online fashion renting, and the potential factors that hinder individuals from participating in online fashion renting. Therefore, it would be worthwhile to look into perceived relative disadvantage and identify the factors that drive such perceptions of online fashion renting in the future. Furthermore, future research can employ different methodological approaches, such as experiment design, to investigate the effects of the specific features of rental platforms on consumers' online fashion renting behaviors.

**Author Contributions:** S.H.L. collected data, developed the literature review, and thoroughly reviewed the manuscript; R.H. analyzed the data and developed the manuscript in the methodology, results, introduction, and discussion. All authors have read and agreed to the published version of the manuscript.

**Funding:** This research received no external funding.

**Conflicts of Interest:** The authors declare no conflict of interest.

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
