# Peer review of "Exploring the Motives for Online Fashion Renting: Insights from Social Retailing to Sustainability"

_sustainability, doi:10.3390/su12187610_

Round 1

Reviewer 1 Report

While the paper generally reads well there are two sentences that would benefit from a re-write: page 1, line 32, 'By using products ....' and page 9, line 299 'Since the study...'

Throughout the paper you use the bracketed reference when discussing the work of others, for example, page 3, line 101, "[8] examined.....". This style is very distracting. It would be better to give the author's surname in addition to the reference number. It is even more distracting throughout page 9.

Page 10, line 350 "....that the both...." delete 'the'

Page 14, line 447, the opening statement you give is not previously indicated and therefore reads rather like an over-statement or even an after-thought. I recommend that you give the reader some indication of this somewhere in the article, perhaps in the introduction, along the lines of the foreseen beneficiaries of this work. 

Author Response

While the paper generally reads well there are two sentences that would benefit from a re-write: page 1, line 32, 'By using products ....' and page 9, line 299 'Since the study...'

 Thank you. We made changes based on your suggestions.

Throughout the paper you use the bracketed reference when discussing the work of others, for example, page 3, line 101, "[8] examined.....". This style is very distracting. It would be better to give the author's surname in addition to the reference number. It is even more distracting throughout page 9.

 Thank you for your comments. The authors’ surnames have been added when appropriate throughout the manuscript.

Page 10, line 350 "....that the both...." delete 'the'

 Thank you. This word been deleted on page 10.

Page 14, line 447, the opening statement you give is not previously indicated and therefore reads rather like an over-statement or even an after-thought. I recommend that you give the reader some indication of this somewhere in the article, perhaps in the introduction, along the lines of the foreseen beneficiaries of this work. 

 Thank you. We have added an explanation about the suggestions of key opinion leaders at the beginning of the paragraph.

Reviewer 2 Report

Thank you for giving me the opportunity to read and review the paper. Online fashion renting plays an important part in fashion sustainability, and more research is definitely needed in this area. However, certain aspects of the paper should be strengthened. 

Introduction:

Lines 51-59  What is your definition of collaborative consumption? Why should the domain-specific type be online fashion renting? You mention that some online fashion renting platforms employ P2P models, but why do you focus on B2C rather than P2P? Many notions are introduced here, but they are not explained clearly. 

Line 70 What does CC refer to? A similar problem occurs in Lines 336 and 358. CFA first appeared in Line 336 but no explanation. 

Lines 76-82 I am wondering whether 300 samples could present a holistic view of extrinsic and intrinsic motivations for online fashion renting. What is the reliability and validity of the survey? Why were there no previous literature on both perspectives?

Literature review:

Line 230 reference? 

Figure 1 shows the final results. At this stage, data results should not be displayed. Why do you need hypotheses if you have already known the results? Figure 1 should be moved to the Discussion section.

Discussion:

The discussion is too simple, and results are not fully discussed. For example, price consciousness did not contribute to relative advantage, so why? (Lines 417-422) More explanations are needed, because it is one of your important findings. 

Author Response

Introduction:

Lines 51-59  What is your definition of collaborative consumption? Why should the domain-specific type be online fashion renting? You mention that some online fashion renting platforms employ P2P models, but why do you focus on B2C rather than P2P? Many notions are introduced here, but they are not explained clearly. 

 Thank you. The definition of collaborative consumption is provided in lines 27-28. Park and Armstrong (2017) argued that the nature of collaborative apparel consumption might be different from that of collaborative consumption in other industry sectors such as automobiles, toys, and/or vacation rentals – the former may meet consumers’ hedonic interests, whereas the latter may satisfy their utilitarian needs. Also, the majority of online fashion renting platforms (e.g., Girl Meets Dress and Meilizu) employ the B2C model, whereas service-based rental market (e.g., Uber, Lyft, and Airbnb) mainly uses a P2P approach. This demonstrates that fashion renting is predominately established in B2C model as compared to P2P. Therefore, this study aims to explore consumers’ motivations for online renting within the apparel domain that emphasizes business-to-customer renting services.

Line 70 What does CC refer to? A similar problem occurs in Lines 336 and 358. CFA first appeared in Line 336 but no explanation. 

 Thank you for the comments. We have spelled out these two abbreviations.

Lines 76-82 I am wondering whether 300 samples could present a holistic view of extrinsic and intrinsic motivations for online fashion renting. What is the reliability and validity of the survey? Why were there no previous literature on both perspectives?

 We followed the rule-of-thumb of “5 or 10 observations per estimated parameter” employed by previous research (see Bentler & Chou, 1987; Bollen, 1989; Kline, 2011). As there are 8 research variables with 30 items in total, 300 samples are enough to present a holistic view to capture extrinsic and intrinsic motivations. Please see the report on the reliability and validity of the constructs in the section on hypothesis testing. Each construct had a composite reliability above .70, and thus both convergent and discriminant validity were confirmed, following the Fornell-Larcker criterion.

Bentler, P. M., & Chou, C. H. (1987). Practical issues in structural modeling. Sociological Methods & Research, 16, 78-117.

Bollen, K. A. (1989). Structural equations with latent variables. New York, NY: John Wiley.

Kline, R. B. (2005). Principles and practice of structural equation modeling. New York: Guilford.

Literature review:

Line 230 reference?

 We have added the number that connects the reference to the list.  

Figure 1 shows the final results. At this stage, data results should not be displayed. Why do you need hypotheses if you have already known the results? Figure 1 should be moved to the Discussion section.

 Thank you for your suggestion. We removed the data result in Figure 1 but indicated the hypotheses. As Table 4 provides the SEM path results, we did not include a figure.

Discussion:

The discussion is too simple, and results are not fully discussed. For example, price consciousness did not contribute to relative advantage, so why? (Lines 417-422) More explanations are needed, because it is one of your important findings. 

 Thank you for your suggestion. We have added more explanation about why price consciousness is not considered as a significant advantage of online fashion renting.

Round 2

Reviewer 2 Report

Thanks for your revision. Accept in present form.

Author Response

The authors thoroughly improved the introduction, methods, and discussion sections based on the reviewers' suggestions. The readability and logic flow are much of progress and empirically sound. A few of the comments leaned on minor changes; thus, I am confident to accept after the minor revision.

A reviewer raised the question of your paper's uniqueness, comparing the previous study of "Tu, J.-C.; Hu, C.-L. A Study on the Factors Affecting Consumers' Willingness to Accept Clothing Rentals. Sustainability 2018, 10, 4139." Tu and Hu had proved the results based on the same theory of TRA and DOI, and the structural equation model. Please justify what originality of the current study has. The authors can add your originality in some places, such as the introduction or Literature review you already cited in their research. In this regard, I might suggest to revise the title not to focus on the theory of TRA and DOI.  For instance, Exploring the Motives for Online Fashion Renting: Insights from Social Retailing to Sustainability ".

 Thank you for your comments. We made change on the title as “Exploring the Motives for Online Fashion Renting: Insights from Social Retailing to Sustainability ".

Also, we justify our study as “Similar but different than Tu and Hu’s study [16], this study tries to investigate how consumers perceive fashion online renting service whether as environmental aspect or fulfilling fashion oriented self- interest” on line 69-71

Moreover, we edit our manuscript twice from professional editors.